# Spatial overlap of sea ice-associated predators and prey in western Hudson Bay

Chloé Warret Rodrigues[1,2]*, Andrew E. Derocher[3], James D. Roth[1], David McGeachy[3,4], Nicholas W. Pilfold[5]

1 Department of Biological Sciences, University of Manitoba, Winnipeg, Manitoba, Canada, 2 Polar Bears International, Winnipeg, Manitoba, Canada, 3 Department of Biological Sciences, University of Alberta, Edmonton, AB, Canada, 4 Environment and Climate Change Canada, Wildlife Research Division, University of Alberta, Edmonton, AB, Canada, 5 Conservation Science Wildlife Health, San Diego Zoo Wildlife Alliance, Escondido, California, United States of America

* warretrc@myumanitoba.ca

## Abstract

Spatio-temporal distribution of species shapes community structure and ecosystem function, yet the mechanisms driving biological hotspots remain unclear in dynamic environments like sea ice. We computed Getis-Ord (Gi*) distribution hotspots based on four years of direct and indirect observations of polar bears (*Ursus maritimus*), Arctic foxes (*Vulpes lagopus*), ringed seals (*Pusa hispida*), and bearded seals (*Erignathus barbatus*) in western Hudson Bay, to identify spatial clustering and assess spatial relationships among these ice-associated species. We further mapped distribution hotspots of bear-hunting sign to examine predator-prey and intraguild relationships. Polar bears and bearded seals primarily used offshore areas, while Arctic foxes concentrated their activity on nearshore ice. Ringed seals built lairs throughout the study area but they mostly hauled out on landfast ice. The polar bear hotspot overlapped largely (30% − 49%) with those of the three other species. Particularly, 80% of the Arctic fox hotspot was included in the polar bear hotspot. In contrast, bearded seals and ringed seals had low overlap (18%), reflecting their different habitat preferences. Understanding current patterns in ice-associated species' distributions and relationships is crucial to inform conservation actions and for predicting direct and indirect effects of Arctic warming. Our results help identify key ecological areas on sea ice and demonstrate how systematic collection of opportunistic observations can be combined to generate valuable ecological insights at low cost.

## Introduction

Spatial and temporal distribution of predators and their prey affect their interactions, driving population dynamics of both predators and prey and thereby shaping community structure and ecosystem functions [1–3]. While predators should preferentially forage in areas of higher prey availability [4,5], prey should avoid high-risk

**Data availability statement:** The data underlying this study is publicly available on Mendeley Data at https://doi.org/10.17632/3pggzffmj8.1.

**Funding:** CWR is grateful for postdoctoral financial support from Polar Bears International (CWR) and MITACS (CWR). Research support was provided by the Banrock Station Environmental Trust (AED), Canadian Association of Zoos and Aquariums (AED), Canadian Wildlife Federation (AED), Environment and Climate Change Canada (AED), Churchill Northern Study Center (CWR), Hauser Bears (AED), Zuest Family Foundation (NWP), Manitoba Agriculture and Resource Development, Natural Sciences and Engineering Research Council of Canada (AED and JDR), Ocelot Grant Program (NWP), Polar Bears International (AED), Polar Continental Shelf Program Canada (AED), Quark Expeditions (AED), San Diego Zoo Wildlife Alliance (AED), and World Wildlife Fund Canada (AED). The funders had no role in study design, data collection and analysis, decision to publish, or preparation of the manuscript.

**Competing interests:** The authors have declared that no competing interests exist.

areas [6,7]. Depending on the outcome of predator-prey interactions, their spatial distribution can correlate either positively (i.e., large spatial overlap) if the predator's response prevails, or negatively (i.e., low spatial overlap) if the prey's response prevails [8].

The stability of such system depends on external constraints, such as spatial anchors, imposed by the need to access specific resources [9]. For example, breeding animals become relatively immobile after establishing dens, lairs, or nests. Predators constrained by spatial anchors are less likely to track prey over large areas, while prey with spatial anchors are more likely to respond to immediate survival threats rather than attempt minimizing encounters with predators by avoiding specific areas [10–12]. Consequently, the expected predator-prey overlap is inherently scale-dependent; at fine spatial scales, prey escaping predators, likely dominate local interactions, whereas at broader scales, predators tend to aggregate in areas of higher prey density, driving broad-scale patterns.

Positive interactions between predators, such as commensalism or mutualism, have recently received recognition as a driving force of population dynamics and animal community structure [13]. In commensal relationships, apex predators can benefit mesopredators by subsidizing them with carrion, with implications for mesopredator population dynamics, food-web structure and stability, and nutrient redistribution between ecosystems [14–17].

Polar bears (*Ursus maritimus*) are apex predators in the Arctic marine food web that depend on sea ice to travel, hunt, and reproduce [18,19]. Where sea ice is seasonal, like in Hudson Bay, polar bears undergo a hyperphagia period in spring — peaking in May and matching the mating, pupping, nursing, and molting periods of their prey — before ice break-up forces them to land where they fast until the ice reforms [20–22]. Polar bears primarily consume ringed seals (*Pusa hispida*) throughout their range, but consume a variety of other prey, including bearded seals (*Erignathus barbatus*), harbor seals (*Phoca vitulina*), beluga whales (*Delphinapterus leucas*), and other marine mammals [19,22–24]. To maximize their energy intake, polar bears primarily consume the blubber of their prey, leaving the rest of the carcass [22,25] available for scavengers [26]. For example, winter sea ice allows terrestrial predators and scavengers, such as snowy owls (*Bubo scandiacus*), ravens (*Corvus corax*), and Arctic foxes (*Vulpes lagopus*), to use marine resources to cope with terrestrial prey scarcity [27–29]. Because Arctic ecosystem productivity is low [30], carrion can be an important food source that supports the reproduction and winter survival of mesopredators [31–33]. However, sea ice use by terrestrial species and spatial relationships between ice-dependent and ice-facultative species are poorly understood.

The Arctic fox is the most well-known scavenger of polar bear kills [26]. In the Nearctic, Arctic foxes primarily feed on lemmings (Arvicolinae) year-round but they forage over sea ice when terrestrial prey are scarce [34–36]. In addition to scavenging on marine mammal carcasses, Arctic fox prey on newborn ringed seals [37,38]. They face little interspecific competition in using marine resources, because their main competitor, the red fox (*Vulpes vulpes*), does not commonly venture onto the

sea ice [39]. Arctic foxes start reproduction in spring and in years of low terrestrial prey availability, marine subsidies enhance reproductive success and increase adult survival, thereby stabilizing population sizes [27,32,40].

The abundant ringed seal and less abundant bearded seal are both broadly sympatric with a circumpolar distribution and peak abundance over continental shelf habitats with shallow waters and seasonal ice cover [41–43]. Ringed seals and bearded seals partition the niche space within the water column and through habitat preferences for hauling out onto the sea ice. Ringed seals primarily forage pelagically and semi-demersally on sea-ice associated prey, whereas bearded seals feed on diverse benthic and pelagic prey [41,44]. Ringed seals maintain breathing holes throughout winter, allowing them to use landfast ice or pack ice [45–47]. Adults select stable consolidated ice with pressure ridges and other ice defor-mations on which snow accumulates and where they build subnivean lairs for protection against predators while resting, birthing, and nursing [21,46,48]. In contrast, bearded seals pup on the surface of sea ice do not usually maintain breathing holes [49–51], and rely on natural sea-ice openings that are prevalent in active pack ice [52–54].

Sea ice provides a platform linking all four species in the marine ecosystem. It is a dynamic environment that under-goes major transformation throughout the year [55], but some features remain stable at coarse spatial scales, such as polynyas and flaw leads, which are formed by winds, currents, and tides. At a finer scale, however, these features vary temporally in shape and size and can modify the icescape on a scale of hours or days [56–58]. For species living in such shifting environments, the temporal variability in resources associated with a given geographical area will likely induce low persistence in their use of a particular area [59].

Aerial surveys have revealed general habitat preferences and variable densities of hauled-out ringed seals and bearded seals [60,61], but fine-scale distributional patterns remain unknown. Because most information comes from individuals hauled-out on top of the ice, little is known about the distribution of ringed seal birth lairs or resting lairs [47,62], with no information available from Hudson Bay. Polar bear space use has been investigated in relation to prey in several parts of its range [47,63,64], but its spatial overlap with scavengers remains largely unexplored. The use of sea ice for foraging and long-range movement by Arctic foxes is well documented [34,35,40,65], yet detailed sea-ice habitat use remains unknown. To date, no study has examined the spatial relationships among polar bears, Arctic foxes, and their pinniped prey.

Our objective was to identify distribution hotspots and spatial relationships among polar bears, Arctic foxes, ringed seals, and bearded seals in the sea ice environment. We hypothesized that both polar bears and Arctic foxes, which face limited interspecific competition, maximize their spatial overlap with resources. We further hypothesized that persistence in space use is low between years, due to the dynamic nature of sea ice. We predicted high spatial overlaps between polar bears and all three other species, a large overlap between Arctic foxes and ringed seals, and a low overlap between Arctic foxes and bearded seals, and between ringed seals and bearded seals reflecting their distinct habitat preferences. Finally, we did not expect high overlap between the yearly hotspots of polar bears and ringed seals based on the temporal dynamics of sea ice.

## Methods

### Study area

The study area encompassed 16,704 km$^2$ of sea ice on western Hudson Bay, primarily to the north and east of Churchill, Manitoba, Canada, extending from ~58.3°N to 59.5°N and 94.3°W to 92.5°W (Fig 1). Hudson Bay is a shallow inland sea characterized by counterclockwise currents and annual ice [66]. Each winter, a major flaw lead forms along the coast of western Hudson Bay (including in the Churchill area), as northwestern winds push the pack ice away from the landfast ice [67]. Sea ice covers >90% of Hudson Bay from December to May, after which it starts breaking up. Sea ice starts to decay in the south and east and the Bay becomes ice-free in July [55]. In comparison to other parts of the Bay, breakup occurs later in western Hudson Bay (including the area near Churchill), and the ice concentration remains high (≥ 5/10) until late in July [55]. Within the boundaries of our study area, the mean ice concentration for the period of our survey (Table 1) was

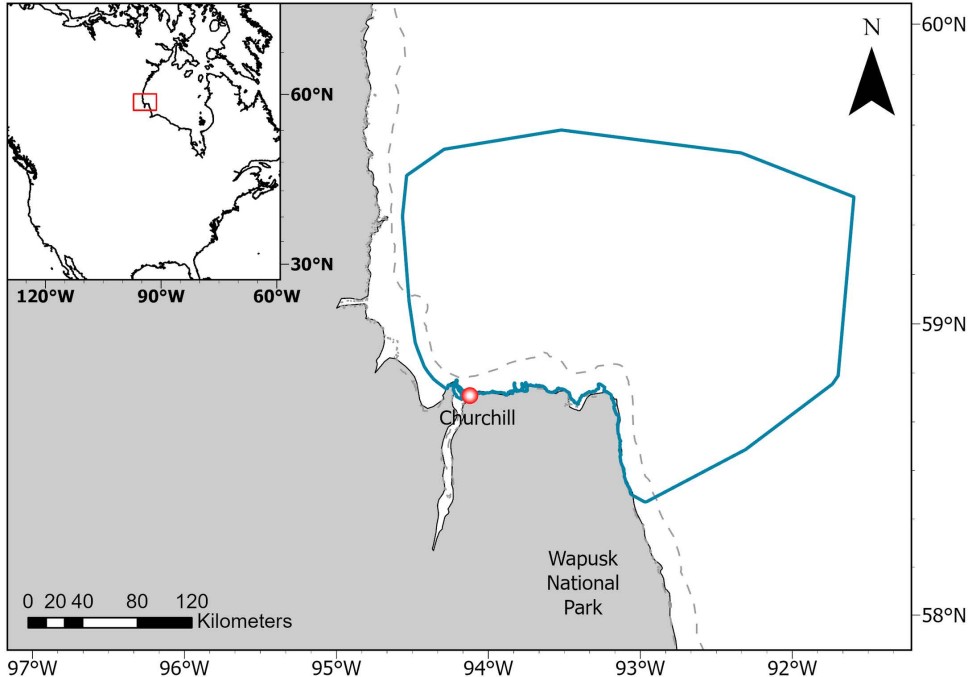

**Fig 1. Study area (blue bounding box) delimited by minimum convex polygon of all helicopter locations pooled over 2019 to 2024.** The grey dashed line represents the typical boundary of the landfast ice.

**Table 1. Sampling effort each year during helicopter surveys for signs of polar bears, seals, and Arctic foxes on the sea ice of western Hudson Bay.**

| Year | Start date | End date | Number of survey days | distance flown (km) | Time flown (hours) |
|------|-----------|----------|----------------------|--------------------|--------------------|
| 2019 | 20-Apr | 30-Apr | 9 | 3753 | 42.3 |
| 2022 | 21-Apr | 05-May | 10 | 5336 | 50.0 |
| 2023 | 25-Apr | 03-May | 9 | 4858 | 45.2 |
| 2024 | 16-Apr | 29-Apr | 11 | 5322 | 51.2 |

97.5% ± 0.4% [50.3% − 100%] (S1 Fig.; calculated using AMSR2-v54 at 3 km resolution), and 50% ice breakup typically occurs between the last week of June and the first half of July (S1 Fig).

## Flight survey

The primary goal of the flight survey was to capture individual bears to fit them with satellite telemetry tags under Species at Risk permits from Manitoba Government SAR24011, SAR23011, SAR21019, SAR18002 and Animal Care permits from University of Alberta BioSciences Animal Care and Use Committee (Animal Use Protocols 00000033 and 00003667). Using a Eurocopter AS350 B2 helicopter, we flew until coming across a bear track, which we would follow until finding the bear or losing its track (S1 Data). During the search, we systematically recorded the coordinates and characteristics of direct observations of ringed seals and bearded seals and spoor of polar bears, and Arctic foxes with CyberTracker (cybertracker.org, Noordhoek, Cape Town, South Africa) on a Samsung A2 tablet (Samsung, South Korea). The surveys were cancelled in 2020 and 2021 due to pandemic restrictions, which resulted in a dataset spanning 4 years (2019, 2022–2024). Flights occurred when cloud cover was minimal, and winds were < 30 km/h. Helicopter location was recorded

every 5 seconds in 2019 and every second in the other years. Typical flight altitude was 75–150 m, and survey duration was 11.25±2.75 days/year [8–14 days] starting on a mean of April 21 [April 16–25] and ending on a mean of May 1 [April 29 – May 5] (Table 1, S1 Table). The number of observers ranged from two to four in addition to the pilot.

Observations recorded in 2019 included seals hauled out on top of the ice (species and number of individuals), polar bear excavations of ringed seal subnivean lairs or snow-covered breathing hole (identified through bear track presence and digging evidence [22]; note that we did not observed subnivean lairs excavated by Arctic fox), and seal kill sites identified by the presence of blood, carcass or remains [68]. For kill sites, we identify seal species when possible and recorded the habitat when distinctive (S2 Table). However, seal carcasses and remains could not always be identified to species level, and we did not have enough carcasses identified to produce a hotspot per species (S2 Table); therefore, all kills were pooled under "seal kills". In 2022–2024, we additionally recorded polar bear and Arctic fox tracks as points, giving each point the coordinates of the initial sighting and treating it as part of the same track until we could no longer follow it.

Based on these observations, we identified six hotspots to reveal patterns of space use and species interactions (Table 2). Hotspots of hauled-out seals (one per species) reflect how ringed and bearded seals used space at the time of the surveys. The ringed seal structure hotspot (hereafter "structure") combines all polar bear digs, whether or not they show clear signs of hunting success (snow fall may sometimes cover blood and small remains, obscuring the signs). Because we could not always identify the type of snow-covered structures and bearded seals can occasionally use or create breathing holes [49,50,69], some structures may have belonged to bearded seals. However, bearded seal's breathing holes are rare compared to ringed seals', and most identified structures are lairs, so any misclassified bearded seal structures are negligible. We thus consider that this layer effectively represents ringed seal space use in winter and spring. The seal kill hotspot serves as a proxy for polar bear hunting focus during their hyperphagic period and provides an assessment of resource distribution

**Table 2. Summary of the layers used for hotspot calculation in this study.** We provide hotspot names used throughout the text, description, type, brief explanation of the ecological insight it provides, and sample size used for hotspot calculation.

| Layer name | definition | type | Ecological insight | sample size |
|---|---|---|---|---|
| **Arctic fox** | Single points representing an Arctic fox track. We assigned to each point the GPS coordinates of initial sighting. A new point was recorded whenever the previous track could no longer be followed. | indirect | Arctic fox space use | 444 |
| **Bearded seal** | GPS coordinates and number of bearded seals observed on top of the ice (i.e., hauled out). Each seal from a cluster received the same GPS coordinates. | direct | distribution of hauled-out bearded seals | 148 |
| **Polar bear** | Single points representing an Polar bear track. We assigned to each point the GPS coordinates of initial sighting. A new point was recorded whenever the previous track could no longer be followed. | indirect | polar bear space use | 2412 |
| **Ringed seal** | GPS coordinates and number of ringed seals observed on top of the ice (i.e., hauled out). Each seal from a cluster received the same GPS coordinates. | direct | distribution of hauled-out ringed seals | 697 |
| **Seal kill** | GPS coordinates of successful polar bear kills, identified by the presence of a carcass or blood. For each kill, we identified the species when possible and recorded whether the kill occurred at a distinctive habitat feature (crack or lead, hole, lair). | direct/ indirect | direct evidence of predator-prey relationship between polar bears and both seal species and indirect evidence of food availability to scavengers | 102 |
| **Structure** | GPS locations of successful and unsuccessful polar bear excavations of ringed seal subnivean lairs or snow-covered breathing holes. We excluded ringed seal carcasses found outside of these features because they may reflect transient or opportunistic use of the ice rather than locations repeatedly used by ringed seals. | indirect | Indirect estimate of ringed seal space use, including during winter and their reproduction | 154 |

for scavenger and direct insight into predator–prey relationships in this system. Finally, polar bear and Arctic fox hotspots, both based on spoor, indicate space use by these two species during the survey period.

## Data standardization

We first thinned helicopter tracks to one location per minute to standardize across years. The survey area was defined by a 3.25 × 3.25 km grid within a bounding box, which we delineated using a minimum convex polygon encompassing all helicopter tracks across years. Because survey flights were regularized line transects, we accounted for uneven spatial survey effort by normalizing species observations per grid cell by dividing the number of observations by the number of helicopter locations + 1 in that cell.

## Hotspot analysis

We conducted hotspot analyses in R v. 4.2 [70] using RStudio v.2024.12.0.467 [71] for each species pooling years using package *sfhotspot* v.0.8.0 [72], and the Getis-Ord Gi* (Gi*) statistic [73]. The hotspot method detects spatial clusters of high (hotspots) and low (coldspots) values by comparing each observation to its surrounding neighbors. We defined neighbors using Queen's case contiguity (i.e., grid cells that share at least one corner or edge). The Gi* statistic assigns a z-score, which indicates whether the observed clustering significantly differs from a random spatial distribution, with positive z-scores representing hotspots and negative z-scores indicating cold spots. As flight paths were non-systematic, our data only support inference on sign presence, not absence. Therefore, we could identify hotspots but not confidently infer coldspots. Each cell's Gi* value was associated with a p-value adjusted using the Holm procedure, which controls the family-wise error rate while remaining more powerful than Bonferroni and more conservative than false discovery rate methods, thereby balancing the risks of false positives and false negatives. We considered hotspots at statistical significance level of $\alpha \leq 0.05$.

To examine spatial relationships among species, we calculated mutual and directional overlaps between hotspots. We imported the hotspots into ArcGIS Pro v. 3.3.0 (ESRI ArcGIS Pro©, Environmental Systems Research Institute Inc., Redland, CA), converted them to polygons, and extracted the contour of significant hotspots. We quantified mutual overlap using:

$$\text{Prop}_{\text{overlap}} = \sqrt{\left(\frac{\text{Area}_{\text{overlap}}}{\text{Area}_A}\right) \times \left(\frac{\text{Area}_{\text{overlap}}}{\text{Area}_B}\right)}$$

(1)

where $\text{Area}_{\text{overlap}}$ is the area shared by the two overlapping hotspots and $\text{Area}_A$ and $\text{Area}_B$ are the individual hotspot areas of species A and B, respectively. We calculated directional overlaps as the area of overlap between two hotspots divided by the area of the hotspot of interest. Mutual overlap was calculated for all species pairs, while directional overlap quantified the proportion of Arctic fox hotspots overlapped by polar bear hotspots, of polar bear and seal hotspots overlapped by seal-kill or structure hotspots, and the proportion of seal-kill, and structure hotspots overlapped by Arctic fox hotspots.

For polar bear tracks (n = 3 years) and ringed seals (n = 4 years), we obtained enough observations to assess temporal hotspot persistence and examine temporal changes in their spatial relationship (Table 3) by measuring distances between yearly hotspot centroids within each species and between species within each year. To ensure consistency of between-year comparisons, we constrained the analysis to hotspot areas within the minimum convex polygon encompassing the region common to all surveys. We quantified yearly hotspot overlap within each species using:

$$\text{Prop}_{\text{overlap}} = \sqrt{\left(\frac{\text{Area}_{\text{overlap}}}{\text{Area}_{Yx}}\right) \times \left(\frac{\text{Area}_{\text{overlap}}}{\text{Area}_{Yy}}\right)}$$

(2)

**Table 3. Raw observation counts per category (fox tracks and bear tracks were not recorded in 2019). 8 seal kills from 2022, 4 from 2023 and 8 from 2024 occurred at lairs or breathing holes and were combined with polar bear digs of lairs and breathing holes to calculate the hotspot "structure".**

|  | 2019 | 2022 | 2023 | 2024 | Total |
|---|---|---|---|---|---|
| Arctic fox track | – | 192 | 207 | 45 | 444 |
| Hauled-out ringed seal | 115 | 217 | 194 | 171 | 697 |
| Hauled-out bearded seal | 28 | 75 | 7 | 38 | 148 |
| Polar bear dig of lairs and breathing holes | 17 | 42 | 36 | 39 | 134 |
| Polar bear track | – | 653 | 970 | 789 | 2412 |
| Seal kill | 6 | 41 | 25 | 30 | 102 |

where $Area_{overlap}$ is the area of overlap between two yearly hotspots, and $Area_{Yx}$ and $Area_{Yy}$ are the hotspot areas in years x and y, respectively. We assessed interannual variation in the overlap between polar bear and ringed seal hotspots by applying Equation 1 separately for each year. Descriptive statistics are provided as mean ± SE.

To assess relationships with sea ice, we calculated the centroid coordinates for each species hotspot and measured centroid distances to shore, and quantified hotspot proportion comprising landfast vs. pack ice. We obtained weekly regional ice data for Hudson Bay from the Canadian Ice Service Digital Archives (https://iceweb1.cis.ec.gc.ca/Archive/page1.xhtml; 2024). To delineate landfast ice, we selected continuous polygons with a 10/10th ice concentration that were attached to the shore [74]. We calculated the maximum extent of landfast ice for each considered period (years pooled or yearly) and quantified the overlap between landfast ice and species' hotspots using a similar equation to (1) and (2). For context, we calculated in ArcGIS Pro v. 3.3.0 the mean percent of the study area covered by landfast ice, and calculated the maximum distance between landfast edge and the coast.

## Results

We surveyed 19,270 km in 188.7 hours across 39 days (Table 1) and recorded 2,412 polar bear tracks, 444 Arctic fox tracks, 697 hauled-out ringed seals, 148 hauled-out bearded seals, 102 seal kills (of which 4 occurred at breathing holes, and 16 at subnivean lairs), and 134 structures (Table 3). Hotspots were identified for all species, structures, and seal kill locations (Fig 2). Landfast ice area represented 5.0% ± 0.2% [4.7% − 6.0%] of the survey area weekly, and fast ice maximum width between the shore and the flaw lead was 19.2 km.

### Species hotspots

Pooled across all years, the ringed seal hotspot was near shore, with a centroid 3.3 km from the coast. It extended mostly west-east, covering 788 km$^2$ (Fig 2). Yearly hotspots varied in proximity to shore, with centroid distances to shore ranging between 5.1 km in 2024 and 16.7 km in 2022 (mean = 9.9 ± 2.9; S1 Table). Yearly-hotspot overlap proportion averaged 30% ± 5%, ranging from 14% (between 2019 and 2022) to 43% (between 2023 and 2024, when hotspots were mostly near shore; S2 Fig). Mean interannual centroid distance was 23.4 ± 4.9 km, ranging from 11.0 km between 2023 and 2024 to 38.2 km between 2019 and 2023 (S2 Fig). The year-pooled hotspot was 70% landfast ice, with yearly values ranging from 26% in 2022 to 81% in 2024.

Pooled across all years, the bearded seal hotspot was mostly located offshore (Fig 2), with a centroid 23.3 km from shore. It covered 940 km$^2$ and extended northward mostly over the pack ice, which represented 95% of the total hotspot area.

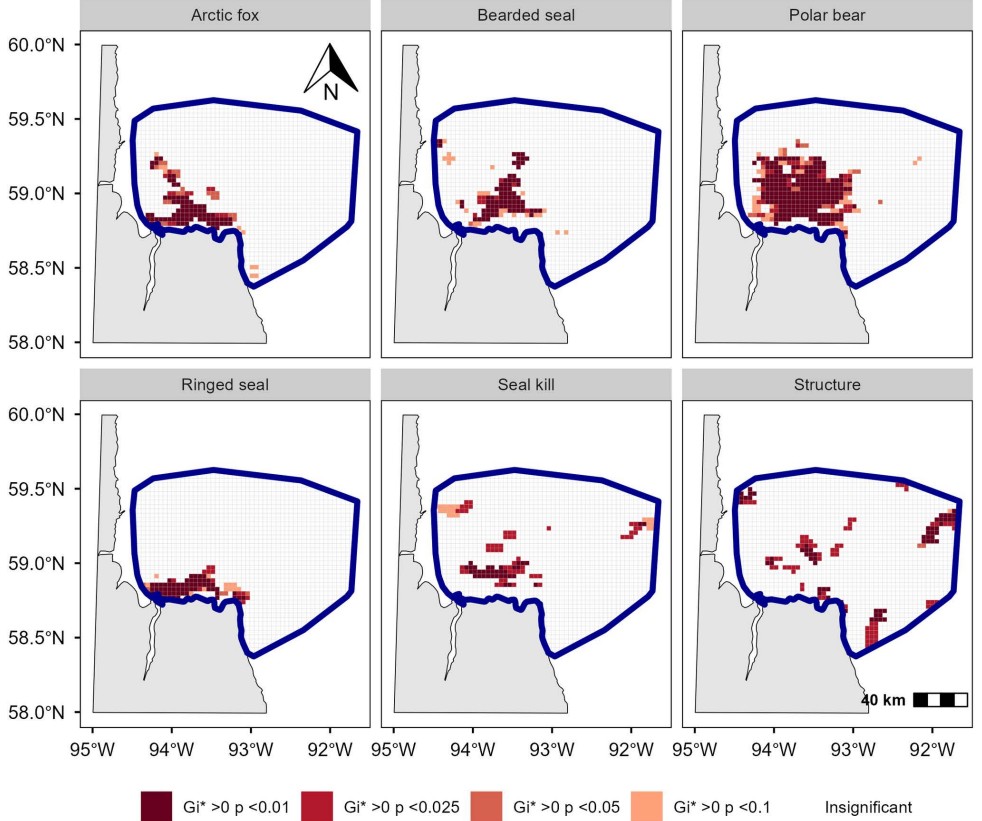

**Fig 2. Maps of the study area in western Hudson Bay with species hotspots and hotspot of events of seal predation by polar bears estimated with the Getis-Ord Gi\* statistics.** Darker levels of red represent increasing levels of statistical significance. Note that we display statistically significant hotspots up to α = 0.1 but only use hotspots statistically significant at α ≤ 0.05 in subsequent spatial analyses.

Pooled across all years, the polar bear hotspot extended over a large portion of the survey area, spanning 2873 km², mostly on pack ice, which represented 88% of the total area (Fig 2) and ranged yearly between 85% (2023) and 92% (2022). The centroid was 25.2 km from shore, with yearly centroid distances to shore ranging from 21.4 to 25.4 km. Yearly hotspot overlap averaged 40% ± 6%, ranging from 28% (between 2022 and 2024) to 48% (between 2022 and 2023). Mean interannual centroid distance was 12.6 ± 2.9 km, ranging from 11.0 km between 2023 and 2024 to 38.2 km between 2019 and 2023 (S3 Fig).

Pooled across all years, the Arctic fox hotspot covered most of the ice near the northern shore but extended onto the pack ice to the northwest (Fig 2). Landfast ice comprised 38% of the total hotspot area, which was 1065 km². The centroid was 14.7 km from shore.

## Species relationships

The largest mutual overlap between year-pooled hotspots occurred between Arctic foxes and ringed seal at 50%, while the smallest mutual overlap occurred between ringed seals and bearded seals at 18% (Table 4). The polar bear hotspot overlapped with all three other species (Fig 3A-C). Polar bears' mutual overlap percentage was 49% with Arctic foxes, 49% with bearded seals, and 30% with ringed seals. Although Arctic fox mutual overlap with polar bears was less than with ringed seals (Fig 3D), 80% of the fox hotspot was included in the polar bear hotspot (directional overlap; Table 4). The Arctic fox and bearded seal hotspots had an intermediate overlap of 31% (Fig 3E).

**Table 4. Hotspot area per species, interspecific area of overlap, and corresponding proportion of mutual or directional overlap. Hotspot area of seal kill and structure overlap are directional (proportion of seal-kill or structure hotspot area overlapping polar bear and seal species hotspot area, and Arctic fox hotspot area overlapping seal-kill or structure hotspot).**

| Hotspot A | Hotspot B | Area A (km2) | Area B (km2) | Area of overlap (km2) | Proportion of overlap | Overlap type |
|---|---|---|---|---|---|---|
| Arctic fox | Ringed seal | 1065 | 788 | 459 | 0.50 | mutual |
| Polar Bear | Arctic fox | 2873 | 1065 | 854 | 0.49 | mutual |
| Polar Bear | Bearded seal | 2873 | 940 | 799 | 0.49 | mutual |
| Arctic fox | Bearded seal | 1065 | 940 | 311 | 0.31 | mutual |
| Polar Bear | Ringed seal | 2873 | 788 | 450 | 0.30 | mutual |
| Bearded seal | Ringed seal | 940 | 788 | 159 | 0.18 | mutual |
| Polar Bear | Arctic fox | 2873 | 1065 | 854 | 0.80 | directional |
| Seal kill | Bearded seal | 898 | 940 | 333 | 0.35 | directional |
| Arctic fox | Seal kill | 1065 | 898 | 316 | 0.35 | directional |
| Seal kill | Polar Bear | 898 | 2873 | 651 | 0.23 | directional |
| Arctic fox | Ringed seal | 898 | 788 | 117 | 0.15 | directional |
| Structure | Bearded seal | 1078 | 940 | 168 | 0.18 | directional |
| Structure | Polar Bear | 1078 | 2873 | 414 | 0.14 | directional |
| Arctic fox | Structure | 1065 | 1078 | 144 | 0.13 | directional |
| Structure | Ringed seal | 1078 | 788 | 70 | 0.09 | directional |

Both the seal-kill and the ringed seal structure hotspots (years pooled) were dispersed, primarily on pack ice, which represented 93% and 88%, respectively, of the total hotspot area (Fig 2). The centroids were 29.6 km (seal kill) and 38.4 km (Structure) from shore. The seal-kill hotspot overlapped 23% of the polar bear hotspot (Fig 3G), only 15% of the ringed seal hotspot (Fig 3H), and 35% of the bearded seal hotspot (Fig 3I). The structure hotspot overlapped 14% of the polar bear hotspot (Fig 3K), 9% of the ringed seal hotspot (Fig 3L), and 18% of the bearded seal hotspot (Fig 3M). The Arctic fox hotspot overlapped 35% of the seal-kill hotspot (Fig 3J) and 13% of the structure hotspot (Fig 3N).

Ringed seal and polar bear hotspot overlap varied interannually with a minimum of 9% in 2023 with a distance between centroids of 27.4 km, and a maximum of 34% in 2022 with a distance between centroids of 12.8 km. The proportional overlap in 2024 was 20% with a distance between centroids of 21.2 km (S3 Fig).

## Discussion

Spatial overlap between predators and prey underpins their interactions, driving changes in demographic rates for both predator and prey populations, which can induce cascading effects in community structure. Using direct and indirect signs of presence, we identified spatial hotspots for four sea ice-associated species. As we predicted, both polar bear and Arctic fox space use matched the distribution of food resources (seals or polar bear kills), while ringed seals and bearded seals showed low spatial overlap. We further found that polar bears had a stronger spatial association with bearded seals, compared to ringed seals. Finally, patterns of sea ice use by these four species were consistent with other studies [20,48,61,62].

Ringed seals mostly hauled out in nearshore areas with some interannual variations. While in 2019 and 2022 their hotspot occurred mainly on pack ice or included large areas offshore, in 2023 and 2024 they concentrated in a narrow band along the shore on the landfast ice. Furthermore, the centroid of the structure hotspot was located 35 km from the coast in pack ice, with minimal overlap of the (hauled-out) ringed seal hotspot (9%). However, lair detection was limited to those excavated by polar bears when they are selecting areas where prey were pupping [75]. We may, thus, have underestimated the use of landfast ice for seal lair construction as this was not a high use area for bears.

Across their range, ringed seals typically prefer shallow waters, relatively high ice cover (40%−80%), stable and consolidated ice, and ice features that accumulate snow to build lairs [37,76−79]. They, however, display

intrapopulation variability in habitat selection [37,41,80]. Notably, they can build birth lairs in the pack ice or give birth in the open when snow accumulation is too low [37,62]. In western Hudson Bay, ringed seals haul out on both landfast and pack ice but occur in lower density on the pack ice [61]. Hudson Bay is a uniformly shallow continental shelf [66], and thus, likely offers abundant suitable ringed seal habitat, unlike the Beaufort Sea, where the bathymetry is more variable and deeper offshore and seals are less likely to be homogeneously distributed [47]. The low overlap between hauled-out ringed seals and ringed seal structures likely reflects a shift in ringed seal behaviors and highlights the use of both pack ice and landfast ice in Hudson Bay, but potentially for different requirements. Notably, the presence of ringed seal birth lairs in the pack ice indicate reproduction on this type of ice, as observed in other Arctic regions [47,60].

In contrast to ringed seals, bearded seals primarily used active pack ice, with their hotspot centroid located farther offshore, consistent with observations across the Arctic [49,81,82]. Consequently, the spatial overlap between hauled-out ringed seals and bearded seals hotspots was low, reflecting their different habitat preferences and a likely shift in ringed

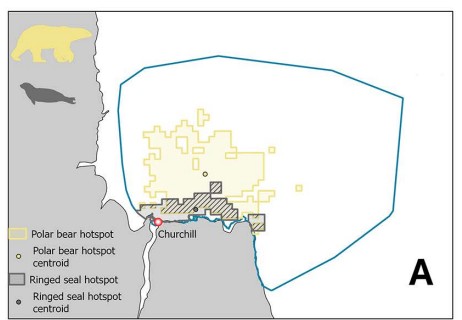
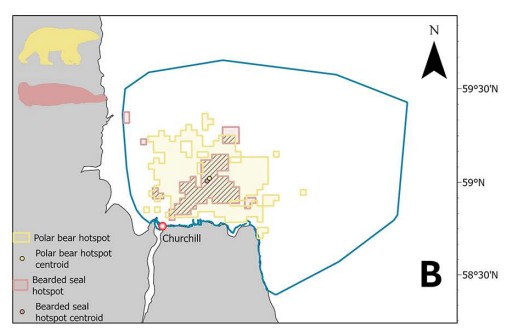
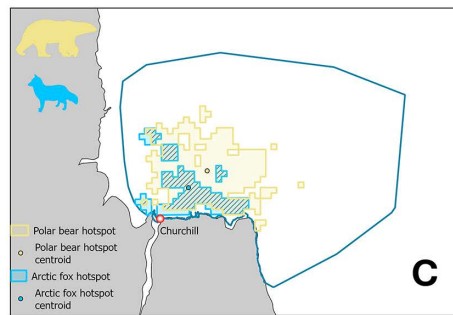
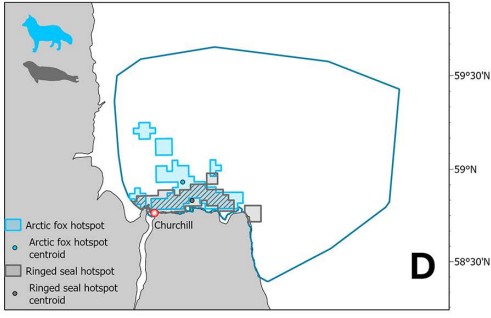
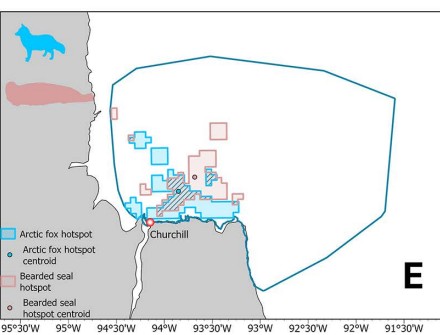
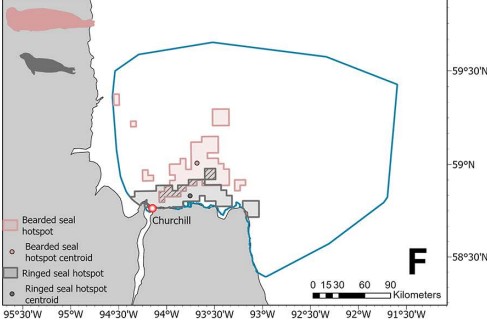

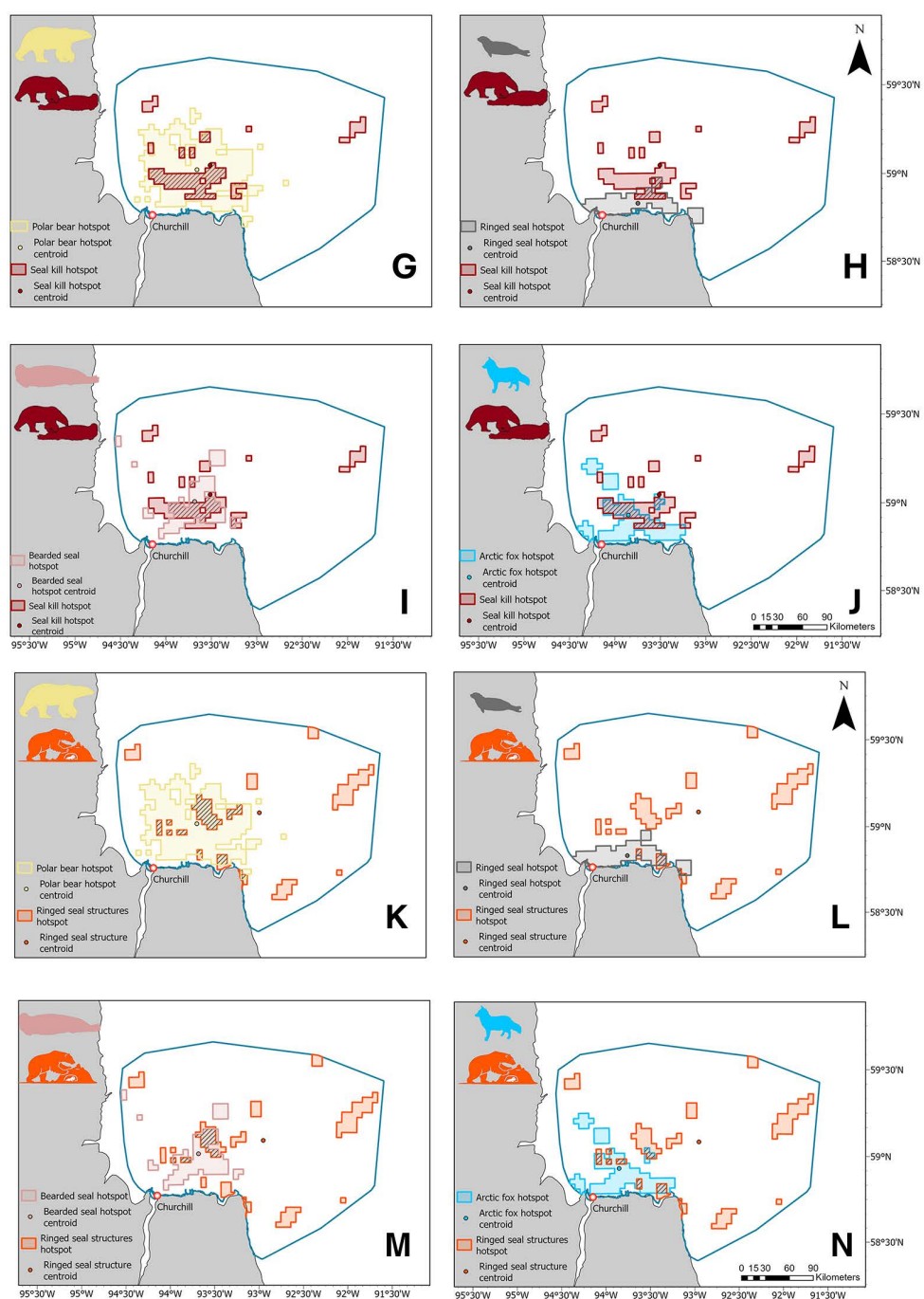

**Fig 3. Hotspots, hotspot centroids, and hotspot overlap between species pairs (Arctic fox in blue, polar bear in yellow, ringed seal in grey, bearded seal in pink, seal kill in red, structure in orange) in the study area near Churchill, Manitoba.** Hotspots were calculated using the Getis-Ord Gi* statistic. We extracted the hotspot area statistically significant at α ≤ 0.05 to produce the overlaps.

seal space use at this time of the year. The presence of a major flaw lead along the coast of the western Hudson Bay enhances area suitability near Churchill for bearded seals [83].

The proportion of landfast ice in polar bear hotspots was generally low but varied considerably between years. In 2022, landfast ice accounted for 8% of the habitat within the polar bear hotspot, a percentage that nearly doubled in 2023. Given the limited variation in landfast-ice coverage within the study area, these shifts likely mirrored the changes in landfast ice proportion within ringed seal hotspots, consistent with their strong predator-prey relationship.

However, despite the strong relationship between polar bears and ringed seals, the low overlap of the structure hotspot with the polar bear hotspot (14%) was unexpected, because ringed seals are the primary prey of polar bears across their range [23,24,84]. Furthermore, the polar bear hotspot overlap with the bearded seal hotspot (49%) was larger than with the ringed seal hotspot (30%), and 35% of the bearded seal hotspot area was covered by the seal-kill hotspot —highlighting the importance of bearded seals for polar bear. These findings suggest that ringed seals were not the sole focus of polar bear predation, possibly indicating behavioral switches from both the polar bear and ringed seals at that time of the year.

In Hudson Bay, ringed seal births end in late April [61]. Most ringed seals may therefore be less constrained by specific habitat features during our study period, as they start to shift away from specific areas used for nursing their young. Without a strong spatial anchor, they could be moving to escape their main predator at coarser spatial scales [9], such as using areas for hauling out that bears do not favor. The flaw lead at the landfast ice edge may attract polar bears due to high prey availability [67,83] but can also be a barrier to access the landfast ice where most ringed seals hauled out, if its width deterred polar bears from swimming across [67]. If bears are less likely to use the land-fast ice, a perceived lower predation risk could have driven ringed seals to use the landfast ice to haul out in the open, as a predator-avoidance strategy. In contrast, bearded seals depend on broken ice, and give birth on the surface of ice near water in late April-May [69]. Only a few days after birth, bearded seal pups are capable of swimming and diving and they spend about half of their time in the water [85]. Their extreme precociality together with a less predictable spatial distribution than ringed seals [86] may induce predator avoidance response at fine spatial scales.

Polar bears select areas that maximize kill biomass rather than kill frequency alone [86]. We found low spatial overlap between polar bears and hauled-out ringed seals, and high overlap between polar bear tracks or hunting activities (i.e., structure and seal kill hotspots) and bearded seals, which suggests that bears prioritized areas with greater access to bearded seals, while also hunting ringed seals. These spatial patterns support earlier findings from the Beaufort Sea [86] and indicate that in spring, polar bears may shift their foraging focus from ringed seals to the energetically more-rewarding bearded seals, especially newly weaned pups that can exceed 100 kg [87]. This strategy may be widely displayed by solitary individuals during the hyperphagic period. Note that we observed few tracks from females with cubs, which suggests that these family groups segregate spatially from the rest of the population and use different foraging strategies, as observed in the Beaufort Sea [86]. By late April to early May, although the peak of ringed seal births has passed, pups remain available, [53], and bearded seals start pupping [82], increasing overall prey availability. Under such conditions, the higher energetic cost of hunting the larger and scarcer bearded seals may become worthwhile to maximize net energy gain, consistent with optimal foraging theory [88,89]. Selecting for pack ice may therefore represent an optimal strategy, granting access to both abundant, more predictable prey with lower energetic value (i.e., ringed seal pups [90]) and scarcer but higher-yield prey (i.e., bearded seals [82,86]). Although prey abundance and catchability are often treated as competing hypotheses, some predators use hunting strategies that account for both [91,92] — a pattern that may also apply to polar bears during hyperphagia.

As predicted, the Arctic fox hotspot overlapped strongly with the polar bear (49%) and seal-kill hotspots (35%), under-scoring their commensal relationship with polar bears and the importance of seals killed by bears as a food source. Arctic foxes also showed high overlap with hauled-out ringed seals (50%). Because our surveys occurred after the peak of ringed seal births, and because bear activity is lower on the landfast ice than on the pack ice, this association may reflect a reluctance of foxes to extend exploratory movements too far from shore or onto the riskier pack ice (if no carcass is

detected). The flaw lead may also act as a barrier at least part of the time, concentrating fox activity along the landfast ice as they search for passage to the pack ice (i.e., where most bear hunting activity occurs).

The low overlap with the structure hotspot suggests that foxes use the sea ice opportunistically when carcasses are available, possibly locating them from considerable distances (up to 40 km at least [93]). Anecdotally, Arctic fox tracks sometimes follow tracks of larger bears, suggesting they may maximize their chances of finding carcasses by cueing on signs from bears with higher hunting success (Derocher, pers. obs.). The delay between the timing of our flights and the occurrence of fox visits to areas searched by bears may also explain this low overlap. Due to constraints related to weather conditions when flying, we may have missed tracks associated with ringed seal structures: snowfall and snow dusting could have obscured Arctic fox tracks, even when larger bear tracks were still visible. The Arctic fox hotspot also overlapped with the bearded seal hotspot more than expected (31%), likely due to the spatial overlap between polar bears and bearded seals. The large size of bearded seal pups at weaning [87] in comparison to ringed seal pups [48,94], likely prevents direct predation by foxes. Instead, foxes' scavenging activity likely creates an indirect spatial association with bearded seals.

Arctic foxes can use sea ice extensively. For example, an Arctic fox collared near Churchill traveled nearly 5,200 km over four months on Hudson Bay's ice (one location per day [35]). Similar long-distance movements across the Arctic [95–97] highlight their reliance on sea ice. Some foxes include large proportions of (landfast) sea ice in their home range during seal pupping season [35]. We found the Arctic fox hotspot centered nearly 15 km offshore and comprised a higher proportion of pack ice than landfast ice. While Arctic foxes' nearshore habitat use may reflect their vulnerability if the flaw lead opens or fidelity to their terrestrial home range, their use of the pack ice highlights their adaptation to navigate this high-risk high-reward habitat.

Studying ice-associated species is challenging due to the inaccessibility of their habitat. While satellite telemetry has significantly advanced our understanding of Arctic food web dynamics, it rarely provides insights on multiple species and their spatial overlap unless each is tracked simultaneously [63,98]. Systematic collection of opportunistic observations of direct and indirect signs of animal presence can provide an alternative for studying rare or cryptic species at low cost [47,99,100]. A limitation of the method is the inability to control detection bias, such as the greater visibility and persistence of larger carcasses [101]. Smaller ringed seal carcasses were likely consumed faster than those of bearded seals, reducing their detectability. Consequently, more polar bear and Arctic fox tracks would be expected to travel to longer-lasting carcasses [26] potentially leading to an underestimation of ringed seal predation. While we advise caution in interpreting our findings, both spatial coverage and intensity were extensive, and the data were standardized. With careful interpretation, we believe that this cost-effective method provides a reasonable representation of true spatial patterns.

## Conservation implications

Understanding how ice-associated species use space and interact is essential for informing targeted conservation. Our value-added survey approach offers valuable insight into species' spatial patterns at a time when only 8.4% of the global ocean (MPAtlas.org) and 5.2% of the Arctic marine area [102] benefit from some level of protection. For example, given the commercial significance of the Churchill harbor, our findings could guide efforts to mitigate the negative effects of increasing human activity by informing the creation of marine protected areas. Species interactions, such as trophic interactions, competition, or pathogen transmission, play a central role in maintaining ecosystem balance. Thus, changes in one species' space use or relative abundance may generate far-reaching effects for both the marine and terrestrial Arctic ecosystems. Our study highlights the importance of considering these dynamics in conservation planning to ensure effective outcomes, particularly in a rapidly changing Arctic.

## Supporting information

**S1 Table. Flight distance (km) and duration (minutes) per ordinal day per year (days in grey were non-survey flights and were not included in the sample effort).**
(DOCX)

**S2 Table. Count of seal carcasses from bearded seals, ringed seals, or unknown species with or without distinctive ice features nearby.** These numbers are provided for context and cannot be used as quantitative evidence: given the nature of the observations, seal availability cannot be estimated, and these raw numbers cannot be converted into interpretable index of hunting effort per seal species.
(DOCX)

**S1 Fig. Mean daily sea ice concentration (%) calculated from AMSR2-v54 at 3 km resolution.** plot (A) shows the daily average for the entire Hudson Bay from 2012 to 2022, with the red line indicating the mean concentration for each ordinal date across all years, plot (B) shows the same metrics for our survey area during the survey period (Table 1, Main Text); the red line again indicates the across-year mean, and the yellow frame highlights the earliest and latest ordinal dates of our surveys. Plot (C) displays the ordinal date of 50% ice breakup for the survey area from 2013 to 2024.
(PNG)

**S2 Fig. Yearly hotspots and hotspot centroids of ringed seal with interannual overlaps (blue 2019 green 2022 purple 2023 yellow 2024).** Hotspots were calculated using the Getis-Ord Gi* statistic within the area common to the 2 years compared. We extracted the hotspot area statistically significant at $\alpha \leq 0.05$ to produce the overlaps.
(PNG)

**S3 Fig. Yearly hotspots and hotspot centroids of polar bears with interannual overlaps green 2022 purple 2023 yellow 2024) Hotspots were calculated using the Getis-Ord Gi* statistic.** We extracted the hotspot area statistically significant at $\alpha \leq 0.05$ to produce the overlaps.
(PNG)

**S4 Fig. Yearly overlap between polar-bear and ringed-seal hotspots (A, B, C). Seals are depicted in grey shades and polar bears in yellow shades, following the convention used in figures from the main text. Hotspots were calculated using the Getis-Ord Gi* statistic.** We extracted the hotspot area statistically significant at $\alpha \leq 0.05$ to produce the overlaps.
(PNG)

**S1 Data. Animated gif showing the helicopter survey paths over the study area accumulating over the four years surveyed (blue 2019 green 2022 purple 2023 yellow 2024).**
(GIF)

## Acknowledgments

We thank Brooke Biddlecombe, Angus Derocher, Sean Headland, Natasha Klappstein, Ryan Mutz, Megan Owen, Nicholas Paroshy, Peter Thompson, Toshio Tsubota, and Yoshiko Torii for their contribution in the field.

## Author contributions

**Conceptualization:** Chloe Warret Rodrigues, Andrew E. Derocher, James D. Roth, David McGeachy, Nicholas W. Pilfold.

**Data curation:** David McGeachy, Nicholas W. Pilfold.

**Formal analysis:** Chloe Warret Rodrigues.

**Funding acquisition:** Chloe Warret Rodrigues, Andrew E. Derocher, James D. Roth, Nicholas W. Pilfold.

**Investigation:** Chloe Warret Rodrigues, Andrew E. Derocher, David McGeachy, Nicholas W. Pilfold.

**Project administration:** Andrew E. Derocher, David McGeachy, Nicholas W. Pilfold.

**Resources:** Andrew E. Derocher, Nicholas W. Pilfold.

**Supervision:** Andrew E. Derocher, Nicholas W. Pilfold.

**Validation:** Chloe Warret Rodrigues.

**Visualization:** Chloe Warret Rodrigues.

**Writing – original draft:** Chloe Warret Rodrigues.

**Writing – review & editing:** Andrew E. Derocher, James D. Roth, David McGeachy, Nicholas W. Pilfold.

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
