## [Decision Letter · Decision Letter 0]

15 Aug 2025

Spatial overlap of sea ice-associated predators and prey in western Hudson Bay

PLOS ONE

Dear Dr. Warret Rodrigues,

Thank you for submitting your manuscript to PLOS ONE. After careful consideration, we feel that it has merit but does not fully meet PLOS ONE’s publication criteria as it currently stands. Therefore, we invite you to submit a revised version of the manuscript that addresses the points raised during the review process.

The reviewer finds the dataset valuable and the manuscript generally enjoyable but notes that the Methods require substantial clarification, justification, and in some cases reconsideration. Key concerns include the definitions and treatment of “hunting attempts” and “kills,” the clarity of survey design and data handling, potential detection biases, and figure/table presentation. Numerous minor wording changes are also suggested for clarity and accuracy.

I also find that the **Methods** section lacks sufficient detail. Much greater clarity is needed on survey effort and design, observation categories, and data processing/analysis. A fuller discussion of detection bias is warranted. In addition, because multiple statistical tests are conducted at α < 0.05, adjustments should be made using an approach such as the Bonferroni correction. All p-values should be reported explicitly, rather than simply stating that they are below a significance threshold.

The current Data Availability Statement may not meet the journal’s requirements; please review the journal’s data availability policy carefully and revise accordingly.

At this stage, I am recommending **Major Revision** . This reflects my view that the manuscript has some chance of eventual acceptance if these issues are fully addressed, though I note that another editor might consider rejecting the current version.

We look forward to receiving your revised manuscript.

Kind regards,

Masami Fujiwara, PhD

Academic Editor

PLOS ONE

[CWR is grateful for postdoctoral financial support from Polar Bears International and MITACS. Research support was provided by the Banrock Station Environmental Trust, Canadian Association of Zoos and Aquariums, Canadian Wildlife Federation, Care for the Wild International, Churchill Northern Studies Centre, Earth Rangers Foundation, Environment and Climate Change Canada, Hauser Bears, Isdell Family Foundation, Offield Family Foundation, Zuest Family Foundation, Kansas City Zoo, Manitoba Agriculture and Resource Development, Natural Sciences and Engineering Research Council of Canada, Parks Canada Agency, Pittsburgh Zoo Conservation Fund, Polar Bears International, Polar Continental Shelf Program of Natural Resources Canada, Quark Expeditions, San Diego Zoo Wildlife Alliance, Schad Foundation, Wildlife Media Inc., and World Wildlife Fund Canada.].

3. In the online submission form, you indicated that [The data will be available upon reasonable request].

5. We note that Figures 1, 2A-F, 3A-F, in your submission contain [map/satellite] images which may be copyrighted. All PLOS content is published under the Creative Commons Attribution License (CC BY 4.0), which means that the manuscript, images, and Supporting Information files will be freely available online, and any third party is permitted to access, download, copy, distribute, and use these materials in any way, even commercially, with proper attribution. For these reasons, we cannot publish previously copyrighted maps or satellite images created using proprietary data, such as Google software (Google Maps, Street View, and Earth). For more information, see our copyright guidelines: http://journals.plos.org/plosone/s/licenses-and-copyright.

1. You may seek permission from the original copyright holder of Figures 1, 2A-F, 3A-F, to publish the content specifically under the CC BY 4.0 license.

6. We notice that your supplementary [figures] are included in the manuscript file. Please remove them and upload them with the file type 'Supporting Information'. Please ensure that each Supporting Information file has a legend listed in the manuscript after the references list.

Additional Editor Comments (if provided):

Reviewers' comments:

Reviewer's Responses to Questions

**Comments to the Author**

1. Is the manuscript technically sound, and do the data support the conclusions?

Reviewer #1: Yes

Reviewer #2: Partly

2. Has the statistical analysis been performed appropriately and rigorously?

Reviewer #1: Yes

Reviewer #2: Yes

3. Have the authors made all data underlying the findings in their manuscript fully available?

Reviewer #1: No

Reviewer #2: No

4. Is the manuscript presented in an intelligible fashion and written in standard English?

Reviewer #1: Yes

Reviewer #2: Yes

Reviewer #1: This manuscript is well written and describes spatial overlap of four ice associated species, which is difficult to achieve. I have edited a Word document in track changes but found few things to comment on. I recommend an additional reference to be considered. Once these minor issues are considered this manuscript should be accepted for publication.

Reviewer #2: The authors used visual observations of polar bears, Arctic foxes, seals, kills, and hunting attempts in Western Hudson Bay to produce hotspots of each observation type and to quantify overlap between them. Information on predator-prey interactions in this ecosystem is limited, making this a valuable dataset. Overall I enjoyed reading this paper, but I felt that some of the methods choices needed significant clarification, justification, and/or reconsideration. A more thorough explanation of what the authors did and why would be extremely helpful. Specific areas of concern are outlined in the attached document along with line-specific comments.

I had to select "no" for the data availability question in the reviewer form because the PLOS One Data Availability section in the manuscript pdf specifically says that "stating 'data available on requestion from the author' is not sufficient" except in an "exceptional situation".

**Do you want your identity to be public for this peer review?** For information about this choice, including consent withdrawal, please see our Privacy Policy

Reviewer #1: **Yes:** Lori Quakenbush

Reviewer #2: No

---

## [Author Response · Author response to Decision Letter 1]

27 Nov 2025

MS-ID: PONE-D-25-36545 “Spatial overlap of sea ice-associated predators and prey in western Hudson Bay”

Academic Editor: Dr. Masami Fujiwara

AE-1: Thank you for submitting your manuscript to PLOS ONE. After careful consideration, we feel that it has merit but does not fully meet PLOS ONE’s publication criteria as it currently stands. Therefore, we invite you to submit a revised version of the manuscript that addresses the points raised during the review process.

Response: We are grateful to the Academic editor for his time and support.

AE-2: The reviewer finds the dataset valuable and the manuscript generally enjoyable but notes that the Methods require substantial clarification, justification, and in some cases reconsideration. Key concerns include the definitions and treatment of “hunting attempts” and “kills,” the clarity of survey design and data handling, potential detection biases, and figure/table presentation. Numerous minor wording changes are also suggested for clarity and accuracy.

I also find that the Methods section lacks sufficient detail. Much greater clarity is needed on survey effort and design, observation categories, and data processing/analysis. A fuller discussion of detection bias is warranted.

Response: We thank the Academic editor for his suggestions. Reviewer 2 identified areas where our methods were not clearly communicated. We have consequently clarified (and justified) our definitions of observation categories, updated several category names to avoid ambiguity, and provided additional details on the statistical analyses for readers unfamiliar with hotspot analyses in R (see response to AE-4 and R2-2). These changes should make the Methods fully transparent and reproducible.

We further fleshed out the discussion of detection bias in the Methods and Discussion.

AE-3: In addition, because multiple statistical tests are conducted at α < 0.05, adjustments should be made using an approach such as the Bonferroni correction.

Response: The hotspot_gistar() function in the sfhotspot package already adjusts p-values using inherited method from R::stats through function p.adjust(). We selected the Holm procedure, which strongly controls the family-wise error rate and is thus more conservative than False Discovery Rate-based methods while being more powerful than the Bonferroni adjustment to detect true effects while keeping false positives under control. By using this method, we are confident that if anything, our results are more at risk to have false negative, rather than false positive. We added a sentence reporting the internal use of the Holm procedure (L.233-236): “Each cell’s Gi* value was associated with a p-value adjusted using the Holm procedure, which effectively controls the family-wise error rate while remaining more powerful than Bonferroni and more conservative than false discovery rate methods, thereby balancing the risks of false positives and false negatives.”

AE-4: All p-values should be reported explicitly, rather than simply stating that they are below a significance threshold.

Response: We can unfortunately not provide exact p-values, because the hotspot analysis produces 1 p-value per cell. The result of the function hotspot_gistar() from package sfhotspot is a tibble with 1 row per cell storing the Gi* value, the associated adjusted p-value and the polygon geometry inherited from package sf. The way hotspots are then extracted relies on a user-defined combination of Gi* value and a selected alpha threshold, such as the code snippet below:

AF_gistar <- df_near %>% #input dataset

st_transform("EPSG:32615") %>% #transform CRS

filter(Observation == "Arctic Fox Track") %>% #filter observation of interest

hotspot_gistar(cell_size = 3250, bandwidth = 8000, weights = n) #compute Gi* cell size ~10Km2 and bandwidth = 8km.

AF_gistar<-AF_gistar %>%

mutate(class = as.factor(case_when(

gistar>0 & pvalue <0.01 ~ "Gi*>0 p<0.01»,

gistar>0 & pvalue <0.025 ~ " Gi*>0 p<0.025 ",

gistar>0 & pvalue <0.05 ~ " Gi*>0 p<0.05",

TRUE ~ "Insignificant"))) #Extraction of hotspot at different alpha threshold

Consequently, we cannot provide a more precise p-value than our selected threshold. Furthermore, while we fully agree that explicitly reporting p-values is important for statistical models such as regressions, in the context of spatial hotspot analysis, reporting the p-value associated with each grid cell would be of limited interpretive value. The Gi* statistic is intended to identify patterns of clustering at a user-defined geographic scale, rather than testing individual cell-level hypotheses. Presenting per-cell p-values would likely overwhelm the reader with inconsequential numbers while obscuring the broader spatial patterns, which are the primary focus of this analysis.

AE-5: The current Data Availability Statement may not meet the journal’s requirements; please review the journal’s data availability policy carefully and revise accordingly.

Response: The data will be stored in Mendeley Data with reserved DOI: 10.17632/3pggzffmj8.1. We apologize for using a placeholder statement during submission while we finalized our choice of repository.

Reviewer 1: Dr. Lori Quakenbush

R1-1: This manuscript is well written and describes spatial overlap of four ice associated species, which is difficult to achieve. I have edited a Word document in track changes but found few things to comment on. I recommend an additional reference to be considered. Once these minor issues are considered this manuscript should be accepted for publication.

Response: We are grateful to the reviewer for the positive comments and suggestions.

We summarize our changes below. Line numbers refer to the original manuscript with corresponding lines in the “Revised Manuscript with Track Changes” between parentheses.

L.24-25: Our sentence read as if the lairs were hauling out, as Reviewer 2 also noticed. We rephrased as follows: “Ringed seals built lairs throughout the study area but they mostly hauled out on landfast ice.” (L.25-26)

L.62-63: All citations converted to numbers and numbers re-ordered to match the order of citation. Thank you.

L.74: “because” added

L.127: “after which” added

L.152: “s” added

L.217: reference to coldspots removed (see response to R2-27 and R2-38). However, we added L.198 that hotspots are clusters of high values, and coldspots clusters of low values.

L.218: We use brackets to indicate ranges because, following mathematical convention, they denote closed intervals (i.e., endpoints are included in the range). We also fixed the space inconsistency when reporting mean ± SE throughout the text (thank you for noticing).

L.244: Indeed, this figure reference should be S2! Corrected. Thank you for catching this typo.

R1-2: L.335-338: But see Rode et al. 2020 DOI: 10.1111/gcb.15572 Adult male bears found to consume more bearded seals than other age and sexes.

Response: Intrapopulation variation in diet has been documented in multiple polar bear populations. In the Chukchi Sea, Rode et al. (2021) indeed reported that adult males consumed higher proportions of bearded seals than other demographic groups, a pattern also observed in earlier studies from other regions (e.g., Thiemann et al. 2007, 2008). In contrast, Galicia et al. (2015) found that adult females in Baffin Bay consumed more bearded seals than adult males. Despite this intrapopulation variability, ringed seals remained the primary prey at the population level in all studies, including Rode et al. (2021). Our statement focused on this population-level trend rather than intrapopulation variation.

We added Rode et al. (2021) to our citations, because it is relevant literature confirming the importance of bearded seals to polar bear diet, and was missing (L.386).

References:

Thiemann GW, Budge SM, Iverson SJ, Stirling I (2007). Unusual fatty acid biomarkers reveal age- and sex-specific foraging in polar bears (Ursus maritimus). Can J Zool 85: 505–517

Thiemann GW, Iverson SJ, Stirling I (2008). Polar bear diets and Arctic marine food webs: insights from fatty acid analysis. Ecol Monogr 78: 591–613

Galicia MP, Thiemann GW, Dyck MG, & Ferguson SH (2015). Characterization of polar bear (Ursus maritimus) diets in the Canadian High Arctic. Polar Bio 38: 1983-1992

L.343: Noted: “still” replaced with “also” to avoid confusion.

R1-3: L.371: This is a ringed seal paper.

Response: We apologize for the confusion. In an earlier version, the statement compared bearded seals to ringed seal pups. We reverted to comparing the two species and separated the citations: “The large size of bearded seal pups at weaning [1] in comparison to ringed seal pups [2,3] likely prevents direct predation by foxes.” (L431-432).

1. Kovacs KM, Krafft BA, Lydersen C. Bearded seal (Erignathus barbatus) birth mass and pup growth in periods with contrasting ice conditions in Svalbard, Norway. Mar Mammal Sci. 2020;36: 276–284. doi:10.1111/mms.12647

2. Hammill MO, Lydersen C, Ryg M, Smith TG. Lactation in the ringed seal (Phoca hispida). Can J Fish Aquat Sci. 1991;48: 2471–2476. doi:10.1139/f91-288

3. Smith TG, Hammill MO, Taugbol G. A review of the developmental, behavioural and physiological adaptations of the ringed seal, Phoca hispida, to life in the Arctic winter. Arctic. 1991;44: 124–131. doi:10.14430/arctic1528

Reviewer 2:

General comments

R2-1: The authors used visual observations of polar bears, Arctic foxes, seals, kills, and hunting attempts in Western Hudson Bay to produce hotspots of each observation type and to quantify overlap between them. Information on predator-prey interactions in this ecosystem is limited, making this a valuable dataset. Overall, I enjoyed reading this paper, but I felt that some of the methods choices needed significant clarification, justification, and/or reconsideration. A more thorough explanation of what the authors did and why would be extremely helpful. Specific areas of concern are outlined in the attached document along with line-specific comments.

I had to select "no" for the data availability question in the reviewer form because the PLOS One Data Availability section in the manuscript pdf specifically says that "stating 'data available on requestion from the author' is not sufficient" except in an "exceptional situation".

Response: We thank the reviewer for the support and for the useful suggestions. We review each comment below.

See response to AE-4: The data will be archived in Mendeley Data with Reserved DOI: 10.17632/3pggzffmj8.1. We changed our statement accordingly.

R2-2: Hunting attempts versus kills: Please clarify whether your “hunting attempts” category also included successful kills. Your description in the Methods section (lines 154-160) and your Figure 2 caption make it sound as though your “hunting attempts” and “kills” categories are mutually exclusive (i.e., hunting attempts do not include kills). Maybe I have misunderstood and your hunting attempt category does include successful kills (which is the case in Pilfold et al. 2014), but if that is the case please clarify. Unless you are specifically interested in UNsuccessful hunting attempts for some reason that is currently not articulated in the paper, it seems to me that the hunting attempts category should include observations of kills. Each kill is inherently a successful hunting attempt, so having a hotspot for all hunting attempts (both successful and unsuccessful) seems like it would be a more ecologically meaningful representation of polar bear foraging effort. I also suspect this would change the location of the hunting attempt hotspot, which currently (and a bit nonsensically) has little overlap with the kill hotspot. To be clear, I do see the value in producing a kill hotspot as you have currently done. But I think analyzing your dataset to produce one hotspot for all hunting attempts (both successful and unsuccessful) and one for kills (a subset of those hunting attempts) would be more valuable than how the observations are currently split.

Keeping with the theme of being confused about the intent behind the “hunting attempts” hotspot… You describe observations of hunting attempts as being based on the presence of polar bear tracks at ringed seal lairs and breathing holes, so I guess it is specific to ringed seals. It would be helpful to be explicit about this, either by saying hunting attempts refer specifically to hunting attempts at ringed seal structures, and/or by adding a sentence clarifying that you don’t expect unsuccessful attempts at catching bearded seals to be detectable.

Additionally, in some places (e.g., line 307-309) you seem to use the hunting attempts hotspot as a proxy for the distribution of ringed seal lairs. However, your Methods text makes it clear that you also counted hunting attempts at breathing holes. Please indicate in the Results what percentage of hunting attempt observations were at lairs versus breathing holes. If your motivation for the hunting attempts hotspot was to have a proxy for ringed seal lairs, then perhaps that category should be limited to observations from lairs only.

Response: We thank the reviewer for the opportunity to reduce confusion in our readers.

The “hunting attempt” and “seal kill” events are mutually exclusive as recorded in the field. A “hunting attempt” is recorded when a polar bear digs into a ringed seal subnivean lair or snow-covered breathing hole and success is not evident (e.g., no carcass, prey remains or blood). If a hunting attempt on ringed seal resulted in an obvious kill, as indicated by the presence of a carcass and blood, then this event is recorded as “seal kill”.

In our analyses, however, the “hunting attempt” layer was a combination of the “hunting attempt” events as recorded in the field and events from “seal kill” directed at structures built by ringed seals (i.e., the hotspot included both unsuccessful and successful attempts to hunt ringed seals). This hotspot was intended as a conservative proxy for ringed seal activity during winter and spring.

To clarify our intentions and Methods section:

1) We renamed the hotspot “hunting attempt” as “ringed seal structure” (or “structure” for short).

2) We clarified the definitions of all observations, particularly hunting attempt (now structure) and seal kill (L.178-191), and added a paragraph explaining why we use each layer (L.192-205).

3) We added a table with each observation term, its definition, whether it is direct or indirect, and the sample size used to calculate each hotspot (Table 2), as an easy-to-consult tool for readers.

4) Table 3 (formerly Table 2) which indicates the raw count of observations may have created further confusion. While most sets of observations directly correspond to a hotspot, the structure hotspots combined all “hunting attempts” and some “seal kills”. We have added the following statement to Table 3 caption to clarify: “Raw observation counts per category (fox tracks and bear tracks were not recorded in 2019). 8 seal kills from 2022, 4 from 2023 and 8 from 2024 occurred at lairs or breathing holes and were combined with polar bear digs of lairs and breathing holes to calculate the hotspot “structure”.”. We also changed “hunting attempt” into self-explanatory “Polar bear dig of lairs and breathing holes”.

5) Hunting attempts are all digging events, but we do not stop at each digging observation; therefore, we cannot always identify the feature that was dug out. Although birth lairs are most likely, male ringed seals can also build resting lairs, and some of these excavations can also be directed at snow-covered breathing holes. We added “snow-covered breathing holes” (L.180) to prevent any confusion.

R2-2: Kills: You note in the Introduction that foxes can kill young ringed seal pups. However, it sounds like all seal kills in your dataset were attributed to polar bears. If this was based on the presence of polar bear tracks at kills, or seal carcasses being too large to have been killed by a fox, it would be good to specify that in the text. Did you have any observations of kills at ringed seal lairs that could have been from foxes? E.g., an excavated lair with blood and no polar bear tracks?

Response: Thank you for drawing our attention to that matter. Indeed,

---

## [Editor Report · Decision Letter 1]

15 Dec 2025

Spatial overlap of sea ice-associated predators and prey in western Hudson Bay

PONE-D-25-36545R1

Dear Dr. Warret Rodrigues,

We’re pleased to inform you that your manuscript has been judged scientifically suitable for publication and will be formally accepted for publication once it meets all outstanding technical requirements.

Within one week, you'll receive an e-mail detailing the required amendments. When these have been addressed, you’ll receive a formal acceptance letter and your manuscript will be scheduled for publication.

Kind regards,

Masami Fujiwara, PhD

Academic Editor

PLOS One
---

## [Editor Report · Acceptance letter]

PONE-D-25-36545R1

PLOS One

Dear Dr. Warret Rodrigues,

I'm pleased to inform you that your manuscript has been deemed suitable for publication in PLOS One. Congratulations! Your manuscript is now being handed over to our production team.

Kind regards,

on behalf of

Dr. Masami Fujiwara

Academic Editor

PLOS One